# AN ATTEMPT TO MODEL HUMAN TRUST WITH REINFORCEMENT LEARNING

## ABSTRACT

Existing works to compute trust as a numerical value mainly rely on ranking, rating or assessments of agents by other agents. However, the concept of trust is manifold, and should not be limited to reputation. Recent research in neuroscience converges with Berg's hypothesis in economics that trust is an encoded function in the human brain. Based on this new assumption, we propose an approach where a trust level is learned by an overlay of any model-free off-policy reinforcement learning algorithm. The main issues were i) to use recent findings on dopaminergic system and reward circuit to simulate trust, ii) to assess our model with reliable and unbiased real life models. In this work, we address these problems by extending Q-Learning to trust evaluation, and comparing our results to a social science case study. Our main contributions are threefold. (1) We model the trust-decision making process with a reinforcement learning algorithm. (2) We propose a dynamic reinforcement of the trust reward inspired by recent findings of neuroscience. (3) We propose a method to explore and exploit the trust space. Experiments show that it is possible to find a set of hyperparameters of our algorithm that reproduces the Dunning-Kruger effect, whereby beginners are overconfident while experts tend to assess their confidence level more accurately.

## 1 INTRODUCTION

Trust seems to be one of the most overused words in the Information Technology world. Yet the concept is struggling to find a universal definition: "the term is never defined. The usage is much clearer than the concept itself" (Hunyadi, 2020). At the end of the day, the question of trust in the digital world is rarely raised; scientists and industrialists prefer to focus on the effects of its erosion - most often - or its improvement - more rarely - on related issues: digital identity, privacy, personal datas, security, customer relationship, artificial intelligence, financial services digitalisation.

On the other hand, from a practical point of view - e.g. for industries and institutions - we observed that there is a lack of tools to forecast the evolution of trust among stakeholders (customers, suppliers, partners) according to the strategic choices of digital transformation. New job positions, such as that of chief trust officer, could be interested in this type of tool. This paper presents the preliminary research that allows us to envision the building of such future applications.

On the understanding of trust as a structure of a social system, we choose to overcome the very first question concerning its definition on the basis of scientific knowledge in sociology, psychology and philosophy.

According to social scientists, trust functions as a way to reduce complexity in a social system, while allowing it to evolve towards richer interactions. It acts as a stabilising structure for societies (Gambetta et al., 2000). Trust makes it possible to act in contexts where information is lacking, or even unavailable (the relations between present and future events being indeterminable). Luhmann (2000) has provided semantic distinction between trust and confidence. *Trust* or *interpersonal trust* is an assessment of risk by an individual who makes a rational choice. Trust allows the binding of groups of people whose number does not exceed 100 to 150 (Aral, 2020; Harari, 2016; Dunbar, 1992) *Confidence* allows an individual to act in spite of an inherent danger in the social system because he has faith in institutionalised factors of the society, his culture, his knowledge. Confidence allows a social system to be unified at scale despite the oppositions of the groups within it.

In his recent work, the philosopher Hunyadi proposes a new unified theory of trust, which leads to the following definition: "Trust is a bet on the behavioural expectations of things (e.g. that the ground will support me when I walk), people (that the driver of the car I pass will obey the rules of the road), and institutions (that the money I use for transactions will have some value)." He argues that former theories - such as Luhmann's - are discontinuous because the intrinsic mechanisms of trust are not the same depending on the scale at which they are observed.

Hunyadi's theory seems both to overlap with Luhmann's definition of interpersonal trust and to be practical enough - e.g. the notion of a bet leads quite naturally to defining a reward in the future - to be simulated with reinforcement learning .

We want to develop a numerical model based on this new definition of trust - the contribution presented in this paper - and hypothesise that such a model would allow the emergence of other types of trust (e.g. Luhmann trust) on a large scale, or even validate the trust continuum hypothesis proposed by Hunyadi - our future works.

The main contribution of this paper is a *trust* learning algorithm based on reinforcement learning. We called this algorithm SelfQ-Trust because the trust level an agent has in itself is evaluated by a Q-value calculated by the algorithm. Leveraging on recent findings about the reward circuit in the human brain, we propose a dynamic reinforcement of the trust reward ; a kind of distributional reinforcement learning applied on a vector of trust levels ; a method for explore and exploit the trust space. We then evaluate the model using a recent dataset of psychological research on the Dunning-Kruger effect on self-confidence.

## 2 RELATED WORK

The economics of internet platforms heavily depend on the quality of information and the security of users (Barbosa et al., 2020). Trust models based on multi-agent systems (MAS) are used to help users choose their interactions by avoiding those who are allegedly incompetent or malicious (Ramchurn et al., 2004; Sabater & Sierra, 2005). Many algorithms assign each agent a score based on the result of its past interactions with others (Jøsang et al., 2007). Both implicit (e.g. frequency, duration, closeness of relationships) and explicit (e.g. comments and ratings from other agents, payment informations) measures are used and the score is considered as a measure of the trust the community can place in the agent (Ben-Naim et al., 2020). The applications cover almost all uses of the Internet. Examples include: e-commerce (e.g. (Xiong & Liu, 2003)), collaborative economy platforms (e.g. Airbnb (Alsheikh et al., 2019)), messaging (e.g. (Lien & Cao, 2014)).

In such a model, the concept of trust is often confused with that of reputation (Jøsang, 2007). To the best of our knowledge, the theoretical frameworks are based on a cognitive model (Cho et al., 2015) or on game theory (Wang et al., 2016). In cognitive models, an agent evaluates the trust it places in another based on the latter's underlying degree of belief. But as it does not have direct access to the mental states of the latter, the trust decision is based on the observation of the outcome of the interaction. The observation model takes into account the uncertainty that arises from the - at least initial - unpredictability of the other agent. In this context, Bayesian networks are a logical choice of knowledge representation (Esfandiari & Chandrasekharan, 2001). In game-theoretic work, trust is the result of a game that involves a utility calculation based on knowledge of the history of relationships. A well-known trust game used in econmics is that of Berg et al. (Berg et al., 1995). The assessment of these models can be done theoretically by establishing a set of properties to be achieved by the system (see e.g. (Cox, 2004)). The experimental approach consists in developing a test bed where different trust systems compete (Sabater & Sierra, 2005).

In economics, Zak & Knack (2001) links trust and economic growth based on game theory. The hypothesis of the rational economic agent is to be reviewed. Indeed, if an agent in an economic system were purely rational, then the trust game of Berg et al. would respect the Nash equilibrium, which is not the case. Several attempts have been made in this direction, the most successful of which use Bayesian models (e.g. (Tribus, 2016)). The stakes are very high. It is a question of taking into account the realistic behaviour of social relations in a society, in order to define - in the long term - theoretical frameworks and indicators capable of making economic decisions in uncertain contexts.

Previous work attempting to measure trust in MAS with Q-learning (see e.g. Vijaya Kumar & Jeyapal (2014); Aref & Tran (2018)) has relied on explicit measures of the relationship - which introduces a modelling bias. Meanwhile, the Berg's hypothesis is being confirmed by recent work in neuroscience. Meyniel et al. (2015) argue that in humans, trust is not derived from a heuristic process but rather is formed during their learning process. For those reasons, we have taken the problem to be that of modelling trust as a human brain learning process by exploiting the latest knowledge on the functioning of the reward circuit.

## 3 BACKGROUND

### 3.1 TERMINOLOGY

We consider a social system $\mathcal{S}$ defined by an environment and a finite set of predefined actions $\mathcal{A}$. A population $\mathcal{I}$ of individuals represented by a MAS whose trust evolution relative to $\mathcal{S}$ is to be modelled. As defined by Russell & Norvig (2020), "an agent is anything that can be viewed as perceiving its environment through sensors and acting upon that environment through effectors". Any agent of $\mathcal{I}$ is defined mathematically as an application $f$ which maps every possible percepts sequence $(o_1, ..., o_n) \in \mathcal{P}^*(\mathcal{O})$, where $\mathcal{O}$ is the set of possible observations of the environment by the agent, to an action $a \in \mathcal{A}$ that the agent can perform or to any other function (e.g., each sub-function of an algorithm intrinsic to the agent) that affects its possible actions: $f : \mathcal{P}^*(\mathcal{O}) \to \mathcal{A}$.

In his doctoral work, Marsh (1992) was the first to formalise trust as a computational concept. He introduced a measure of trust $T_x(y, \alpha) \in [-1, 1[$ as the probability that an agent $y$ will act if $T_x(y, \alpha) \geq 0$ (respectively minus the probability it will not act, if $T_x(y, \alpha) < 0$) in the situation $\alpha$ to reach an outcome that another agent $x$ presupposes to expect if it had a full trust in $y$, before $y$ acts. If $T_x(y, \alpha) > threshold(y, \alpha)$ the agents cooperate.

Research in social sciences and information technology has extended the English terminology from property law [1] [2]. Thus, according to Hardin (2002), a "trustor" is an entity that trusts another entity, the "trustee". According to Cofta (2007), a *trustor* can be a social actor (such as a person or an institution) or a technical actor (such as a computer or a software), which acts on behalf of a social actor. In the following, we will note $f \in \mathcal{I}$ to refer to a trustor who is placing its trust in a trustee $g \in \mathcal{I}$ when it wants to perform an action chosen from a part of the available actions $\mathcal{P}(\mathcal{A})$.

### 3.2 Q-LEARNING

A single-agent reinforcement learning problem is commonly modelled by a Markov decision process (MDP). A MDP is usually formalized by a tuple $\{\mathcal{S}, \mathcal{A}, T, \mathcal{R}\}$ where $s \in \mathcal{S}$, $a \in \mathcal{A}$ and $r \in \mathcal{R}$ stands respectively for state, action and reward. $T$ is a transition function defined as a probability distribution over the states. For any $s \in \mathcal{S}$, agent's choice to perfom an action $a \in \mathcal{A}$ occurs with the probablility $T(s, a, s') \in [0, 1]$ where $s'$ is the next state following $s$. It will result in the environment entering the new state $s'$ and give a reward $r = R(s, a)$. A policy $\pi$ describes which action the agent takes in each state. Formally, it is therefore a function $\mathcal{S} \to \mathcal{A}$ in the case of a deterministic policy or $\mathcal{S} \times \mathcal{A} \to [0; 1]$ in the stochastic case. The goal for the agent is to learn a policy that maximizes the cumulative rewards received over its learning process.

A common way to achieve the goal is to solve the Bellman equation applied to the total sum of discounted rewards $Q_\pi(s, a)$ reaped by the agent if it starts in state $s$ and first takes the action $a$, before then applying the policy $\pi$ ad infinitum ($\gamma$ states for the discount factor):

$$q_*(s, a) = \sum_{s', r} p(s', r | s, a) \left[ r + \gamma \max_{a' \in \mathcal{A}} q_*(s', a') \right] \tag{1}$$

Introduced by Watkins (1989), Q-learning is a numerical method of solving Equation 1. At each time step $t$ of the algorithm, agent chooses action $A_t$ from state $S_t$ using policy derived from $Q$ (e.g.

---

[1] **trustor** (*Collins*): (in property law) a person who sets up a trust transferring property to another person
[2] **trustee** (*Collins*): someone with legal control of money or property that is kept or invested for another person, company, or organization

$\epsilon$-greedy). Then it makes a step in environment, observe the new state $S_{t+1}$ and the reward $R_{t+1}$. Q-value is updated by applying a temporal-difference method (TD) (Sutton, 1988) :

$$Q(S_t, A_t) \leftarrow Q(S_t, A_t) + \alpha \left[ R_{t+1} + \gamma \max_{a \in \mathcal{A}} Q(S_{t+1}, a) - Q(S_t, A_t) \right] \qquad (2)$$

where $Q$ is the learned action-value function which approximates the optimal action-value function $q_*$ (Watkins & Dayan, 1992), independent of the policy being followed and $\alpha \in [0, 1]$ is a learning rate setting the updating speed of Q-values at each time step. The part of the equation in brackets is the difference between the reward estimated by the model and the reward actually received.

### 3.3 Modelling the Dopaminergic System

Berg hypothesised that trust is a function that has been encoded in the human brain. Cognitive neuroscientists linked the functioning of the neurotransmitter dopamine to the theoretical computational framework of reinforcement learning in the early 2000s (Holroyd & Coles, 2002). It is now established that the release of dopamine in the reward circuit reveals a prediction error by the individual (Glimcher, 2011). This surprise signal favours the learning of reward predictions and shapes the individual's future behaviour.

Predictions of reward were previously represented as a single scalar quantity. However, recent artificial intelligence research on distributional reinforcement learning has inspired a new model in which the brain represents possible future rewards not as a single mean, but rather as a probability distribution, effectively representing multiple future outcomes simultaneously and in parallel (Bellemare et al., 2017). Recordings from the mouse ventral tegmental area provided strong evidence for the neural realisation of distributional reinforcement learning (Dabney et al., 2020). From a functional point of view, an interesting recent hypothesis suggests that dopaminergic neurons encode both differences between rewards and expectations in a goal-directed system. These prediction errors trigger reward learning (Bogacz, 2020).

## 4 Methods

We propose the following definition based on Hunyadi's work:

**Definition 1 (Trust)** *Trust is a bet made by an intelligent agent on the behavioural expectations it has of another agent or of itself. Considering that an agent $f$, named trustor, trusts another agent $g$, named trustee, we define a level of trust $T_f(g) \in [-1, 1]$ as a measure of the subjective certainty of the trustor $f$ that its bet will come true if the level is positive, and false if the level is negative.*

### 4.1 Problem Statement

We consider a population $\mathcal{P}$ of agents whose behaviour in a given environment is modelled by a MDP $\{\mathcal{S}, \mathcal{A}, \mathcal{R}\}$. The agents learn an optimal policy by means of any underlying model-free off-policy reinforcement learning algorithm.

The goal is to estimate the trust level $T_f(g, s) \in [-1, 1]$ that a trustor $f \in \mathcal{P}$ can have in any agent $g \in \mathcal{P}$ which advises it to take the action $a \in \mathcal{A}$ in state $s \in \mathcal{S}$ following the introspection of its MDP model. We require that: (1) the trust measurement algorithm runs in parallel with the MDP learning algorithm and (2) it can be implemented with any model-free off-policy RL algorithm.

### 4.2 Learning Principles

The general explanations concern the multi-agent case, in which each agent assumes both the role of *trustor* and that of *trustee*. For a better understanding of the basics of the method, simplifying assumptions are made: the trust environment perceived by the agent is deterministic and the *trust actions* defined hereafter are discretised.

The trust level $T_f(g, s)$ is learned by a trust model formally described by a MDP $\{\mathcal{S}, \hat{\mathcal{A}}, \hat{\mathcal{R}}_{f,g}\}$, where $\hat{\mathcal{A}} = [\![0, 2n]\!]$ is the set of trust actions corresponding to trust levels $\{\frac{\hat{a}-n}{n}\}_{0 \leq \hat{a} \leq 2n}$ and $\hat{\mathcal{R}}_{f,g}$ :

$S \times \hat{A} \to \mathbb{R}$ is a reward function. By applying the Definition 1, we define the trust action $\hat{a}$ as follows: $f$ trusts $g$ to take action $a$ chosen by $g$ w.r.t. its underlying MDP model with trust level $T = \frac{\hat{a} - n}{n}$ means that the closer $T$ is to 1, the greater $f$ is certain that action $a$ is optimal regarding the underlying MDP and the closer $T$ is to $-1$, the greater $f$ is certain that action $a$ isn't optimal regarding the underlying MDP.

**Overall operation** The trustor $f$ asks any agent $g$ for the action $a$ it would have taken if it had been in the same state $s$ as $f$. This advice allows $f$ to choose its action. $f$ then learns its own model of trust towards $g$ thanks to a trust reward $\hat{r}$. $g$ may be a trustee found in a queue of trusted agents or $f$ itself. In the latter case, the algorithm calculates the trust that $f$ has in itself.

To do so, the algorithm learns a Q-function $\hat{Q}_{f,g} : S \times \hat{A} \to \mathbb{R}$ that calculates the quality of a state–trust action combination. At all times, the trust that $f$ places in $g$ in state $s$ can be derived from the Q-function. Equation 3 is the application of Q-learning in the state-trust actions space. Applying the Q-learning principle, the trust level estimated by the model is therefore the one obtained thanks to the index of the trust action for the max of the Q-value. To report it in the interval $[-1, 1]$, we divide by $n$ and subtract 1.

$$T_f(g, s) = \frac{\arg\max_k q_k - 1}{n} - 1$$
$$\text{with} \quad \hat{Q}_{f,g}(s) = (q_k)_{1 \leq k \leq 2n+1} \text{ the Q-value trust vector} \tag{3}$$

However, our initial attempts to make a simple instantiation of Q-learning in the state-trust action space yielded poor results. We faced two main challenges: (1) a fixed reward did not work for learning a realistic trust model. (2) when reinforcement is applied to only one trust level, the model quickly locks on the initial level. We used the following approach to answer these two issues.

**Using the neural realization of distributional reinforcement learning hypothesis.**

Firstly, we hypothesised that trust is a process that is built up in the human brain. Secondly, we define this process as a bet - therefore fundamentally reward-based. We therefore require that our model takes into account the latest neuroscientific knowledge about the reward circuit in the human brain. The reward prediction error (RPE) hypothesis of dopamine — the discrepancy between observed and expected reward — has become central to research at the intersection of neuroscience and computer science because RPE is precisely the signal that a RL system would need to update reward expectations (Montague et al., 1996; Schultz et al., 1997). Dabney et al. findings provide strong evidence to support the hypothesis that the brain represents future rewards not as a single scalar quantity, but rather as a probability distribution.

We apply the hypothesis to our model of trust learning. We therefore require that (1) the reward magnitude must be updated according to an observation of the result of the bet ("Dynamic reinforcement" section below) and (2) the bet made by the trustor in a trust relationship with a trustee must give rise to several simultaneous rewards over all possible trust levels ("Distributional reinforcement learning" section below).

**Dynamic reinforcement.** A way to vary the reward according to the experience of trust relationships were to introduce a kind of attention mechanism. The general idea is (1) to increase the reward at the end of an episode when a positive trust action leads to an improved score and conversely (2) to make the reward proportional to the trust level of the trust action. For this purpose, we assume that the agent is able to measure its performance and that of other agents in solving the underlying MDP. The trustor uses this measure and the prior trust level it has towards its advisor to calculate the magnitude of the trust reward. The magnitude of the trust reward $\hat{m}_{f,g}$ is regularly updated according to the observed performance of the trustee $g$ in finding the optimal policy of the underlying MDP. Each time the trustor $f$ chooses an action advised by a trustee $g$, it receives a trust reward $\hat{r}$ depending on the prior trust level $T$ it places in $g$ and the magnitude of the trust reward $\hat{m}_{f,g}$:

$$\hat{r} = \hat{m}_{f,g} T \tag{4}$$

**Distributional reinforcement learning.** We tried to apply Equation 2 to the Q-value of trust level $T$ prior to the relationship and observed that the learned trust level quickly locks onto its initial level. We chose to use a distributed learning design to overcome this problem, while taking advantage of the

distributed reinforcement brain hypothesis. We therefore apply the learning to all possible Q-values of trust actions at the given state-action pair. The $2n$ trust levels $T_i$ are learned by $2n$ Q-functions $\hat{Q}_i$ simultaneously and in parallel, as shown in Figure 1.

In the following, $T$ is the *a priori* level of trust that the trustee $f$ has in the trustee $g$. At each time step of the algorithm, a difference $\delta(\hat{\alpha}, \hat{\gamma}, \hat{r})$ between the reward estimated by the model and the reward actually received $\hat{r}$ is computed in the same way as in Equation 2. A Gaussian random variable $X$ centered on $T$ is modulated by this difference. Equation 5 is used to update the $2n$ functions $\hat{Q}_i$ corresponding to the $2n$ discrete trust levels:

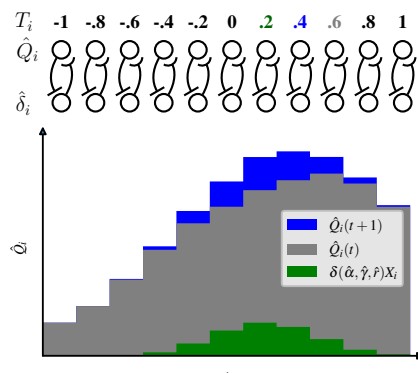

$$\hat{Q}_{f,g}(S_t) \leftarrow \hat{Q}_{f,g}(S_t) + \delta(\hat{\alpha}, \hat{\gamma}, \hat{r})X$$
$$\text{with} \quad X \in \mathbb{R}^{4n+1} \sim \mathcal{N}(T, 1) \tag{5}$$

**Exploration and exploitation of trust model**. The trustor has to manage a trade-off between trusting other agents without prior knowledge (trust exploration) and using the trust model it has learned to take into account other agents' advices (trust exploitation). We use for that an epsilon-greedy method where: (1) Trust exploration amounts to the trustor choosing to have full trust in the trustee; we hypothetize that trusting someone - or oneself - in a first relational experience - should it be self-oriented - is choosing to systematically take into account the advice he/she gives. (2) Rather than just using the model being learned, the exploitation involves an uncertainty on the level of trust that the trustor grants to the trustee. In fact, as it is stated in the survey of Cho et al., the uncertainty on the a priori trust that one grants in an interpersonal relationship tends to fade with time. In the single-agent case, only the phase of exploitation is played.

Figure 1: Distributional trust learning. In the example, before the trust action, the trust level of the trustor towards the trustee is $0.6$, as given by the model in gray. The trustor chooses an *a priori* trust level in range $[0.6 - \hat{\epsilon}, 0.6 + \hat{\epsilon}]$ - say $0.2$ (see Equation 7 in next section). A Gaussian random variable $X$ centered on $0.2$ is modulated by the difference between actual reward and expectation (in green). Added to the Q-value, the result gives the model after the trust action in gray, indicating that the trust level drops from $0.6$ to $0.4$.

### 4.3 SINGLE-AGENT CASE

SelfQ-Trust Algorithm (see Appendix A) focuses on the single-agent case, with agent $f$ assuming both roles of trustor and trustee. From a technical point of view, the trust evaluation should be seen as an introspection function of the agent that evaluates in real time the relevance of its learning algorithm's prediction of an underlying MDP with respect to its current state. In the single-agent case, the evaluation of trust amounts to the evaluation of self-confidence. For better clarity, we instantiate below Definition 1 on the single-agent case.

**Definition 2 (Self-confidence)** *Self-confidence is a bet made by an intelligent agent on its future action. We define a level of self-confidence $T_f \in [-1, 1]$ as a measure of the subjective certainty of the agent $f$ that future action it will take is optimal with respect to the task it is currently solving if the level is positive, and non optimal if the level is negative.*

For the dynamic reinforcement, we update $\hat{m}_{f,f}$ at the end of each episode $i$ w.r.t. the score $v_i$ the agent achieved in the episode $i$. Equation 6 is a heuristic that weights the reward of the trust model according to the score that the agent measures at the end of a learning episode of the underlying MDP. The idea is as follows: we consider that the agent stores its scores in a $memory$ list of fixed-size. If the agent beats the highest score it has stored in the last $card(memory)$ episodes, one can consider that it was right to trust itself over the past episode, driving to a positive reward for the next episode. Conversely, if the agent does less well than its highest score, one can consider that it was wrong to trust itself on the past episode, driving to a negative reward for the next episode. The magnitude of the reward will be proportional to the difference with the highest score observed in the memory.

$$\hat{m}_{f,f} = -\frac{v_i - \max\limits_{v_j \in memory}(v_j)}{v_i} \tag{6}$$

For the exploitation of trust model, Equation 7 allows for uncertainty in the level of trust $T$ that the agent grants to itself. The uncertainty is decreasing with time, i.e. it is depending on the probability $\hat{\epsilon}$ of choosing to explore the trust model.

$$T = \frac{X}{n} \text{ , with } X \sim \mathcal{U}\left(\max(-n, n(T_f(f,s) - \hat{\epsilon})), \min(n, n(T_f(f,s) + \hat{\epsilon}))\right) \tag{7}$$

## 5 EXPERIMENTS

There are public quantitative surveys dealing with trust (e.g. interpersonal trust, confidence in public institutions (Fitzgerald et al., 2016; Ortiz-Ospina & Roser, 2016)). Nevertheless, the available datasets are likely to be extremely biased the term is never defined. Moreover, the interpretation of general questions about trust can be very different from one individual to another.

However, there are studies in the social sciences and economics that give results on observable properties of known systems. These studies can be used as test cases to assess the model. Among them, we have chosen the so-called *Dunning-Kruger effect* (DKE) to assess the trust of an agent in itself. Specifically, in their seminal paper, Kruger & Dunning (1999) put forward as an explanation a metacognitive difficulty of the unqualified people which prevents them from accurately recognising their incompetence and evaluating their real abilities. Conversely, the most qualified people would tend to underestimate their level of competence and would wrongly think that tasks that are easy for them are also easy for others. Dunning (2011) suggests that the effect is universal, as it has been observed in very different areas of learning.

**Experimental Setup**. We implemented the SelfQ-Trust algorithm in Python; the code is available at `https://github.com/selfQtrust/code`. Environment is a randomly generated maze of dimension $(n, n)$. The goal of the agent is to start from the upper left corner and find the exit at the lower right corner. The maze generation algorithm ensures that at least one path exists between the entrance and the exit.

The agent learns to trust itself to select the optimal path in each state of the environment. In the maze case, the state is the cell where the agent is located. Thus, during an episode, an agent learns a sequence of trust levels, each of which is linked to a cell he visited. Using this sequence, we calculate three metrics to aggregate the trust measures during the episode: the mean trust on all states of the environment at the end of the episode ($M1$); the mean trust over all states the agent passed through during the episode ($M2$); the max trust the agent had during the episode in the states it passed through ($M3$).

**Simulation of the Dunning-Kruger Effect (DKE).** In our first experiments, we were looking at whether the algorithm is able to reproduce the DKE effect. We trained a population of 1000 agents over 3 mazes of dimension $(6, 6)$, $(10, 10)$, $(30, 30)$. Computations of the three metrics $M1$, $M2$, $M3$ are shown in Figure 2.

The results show that the SelfQ-Trust algorithm reproduces the DKE trend: the agent's self-confidence goes through a peak systematically present at the beginning of the training, then decreases abruptly to a more or less pronounced local minimum, to end up rising and then stabilising. Moreover, the effect strength seems to increase with the complexity of the environment. The following experiment raises the question of the dependence of DKE on hyperparameters.

**Sensitivity to learning hyperparameters.** The learning model uses two pairs of learning hyperparameters: the underlying MDP that the agent uses to solve the maze problem uses an $\alpha$ learning rate and a $\gamma$ discount factor; the trust model uses a rate of trust learning $\hat{\alpha}$ and a discount factor of trust learning $\hat{\gamma}$. The environment is the same maze of dimension $(5, 5)$ for the whole experiment. We trained a population of 50 agents on 60 episodes of MDP learning as well as on 60 episodes of trust learning - we choosed those settings accordingly to Sanchez & Dunning (2018) experiments on Overconfidence Effect. The second metric $M2$ is used to show the results in Figure 3.

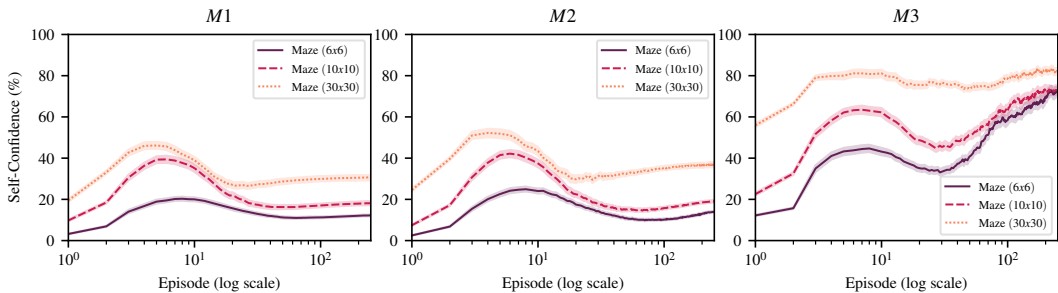

Figure 2: DKE simulation. 1000 agents are trained on 500 synchronised episodes of the underlying MDP and the trust model on each of the 3 mazes. Hyperparameters are set to: $(\alpha, \gamma, \hat{\alpha}, \hat{\gamma}) = (0.1, 0.99, 0.01, 0.99)$ for the 3 mazes. Estimate of the central tendency (mean estimator) and 95% confidence interval of self-confidence computed by SelfQ-Trust vs. learning episode.

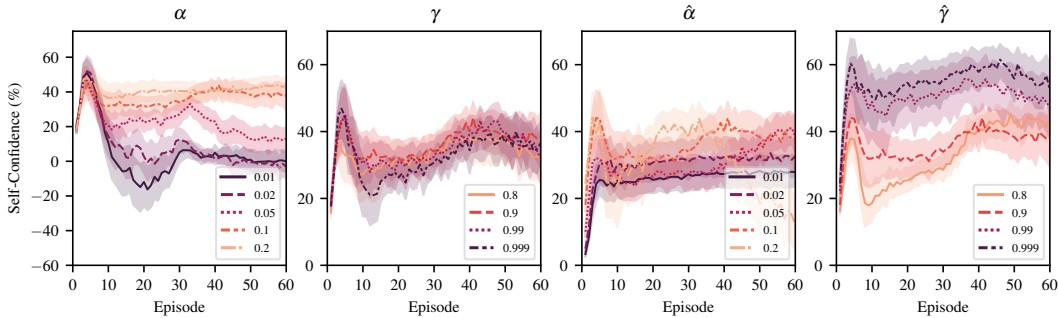

Figure 3: Sensitivity to learning hyperparameters $\alpha$, $\gamma$, $\hat{\alpha}$ and $\hat{\gamma}$. We set $(\alpha, \gamma, \hat{\alpha}, \hat{\gamma}) = (0.1, 0.9, 0.1, 0.9)$ then for each of the figures we vary the hyperparameter studied in the set $\{0.01, 0.02, 0.05, 0.1, 0.2\}$ for $\alpha$ and $\hat{\alpha}$ and in the set $\{0.8, 0.9, 0.99, 0.999\}$ for $\gamma$ and $\hat{\gamma}$. Estimate of the central tendency (mean estimator) and 95% confidence interval of self-confidence computed by SelfQ-Trust vs. learning episode.

The experiment confirms the DKE trend, regardless of the hyperparameters choosen. Moreover, the magnitude of the trend appears to be correlated with $\alpha$, $\hat{\alpha}$ and $\hat{\gamma}$ on the experience. $\gamma$ does not seem to have much effect on the trust model (all curves are in the same statistical confidence interval). We wonder why the peak of confidence - which we will later call the peak of overconfidence - almost always occurs during the same short window of learning episodes?

**Sensitivity to the complexity of the environment.** The complexity of the environment should be narrowly assessed from the agent's point of view. We thus calculate it by training a population of agents with the underlying MDP learning algorithm - i.e. a simple Q-learning on a maze environment in the case of the experiment. We randomly generate 1000 mazes of dimension $(10, 10)$. We train an agent on each over 1000 episodes with $\alpha = 0.1$ and $\gamma = 0.99$. We define complexity as the smallest number of steps the agent has taken in the environment during the training episodes.

We then train 600 agents with SelfQ-Trust on a subset of mazes according to the range of the measured complexity values. The number of episodes of the underlying MDP training and that of the trust model are equal to 60. Hyperparameters are set to: $\alpha = 0.1$, $\gamma = 0.99$, $\hat{\alpha} = 0.1$, $\hat{\gamma} \in \{0.9, 0.99\}$. Results are in Figure 4.

Figure 4(c) show that the beginner's overall self-confidence tends to increase with the complexity of the task being learned. It may be hypothesised that, the more complicated the problem is to solve, the more difficult it will be to learn self-confidence and the more biases in that learning - such as overconfidence - will manifest themselves. Moreover, the closer the trust discount factor is to 1, the greater the overconfidence effect. Shall the $\hat{\gamma}$ factor of SelfQ-Trust algorithm be able to measure the overconfidence effect studied by Sanchez & Dunning?

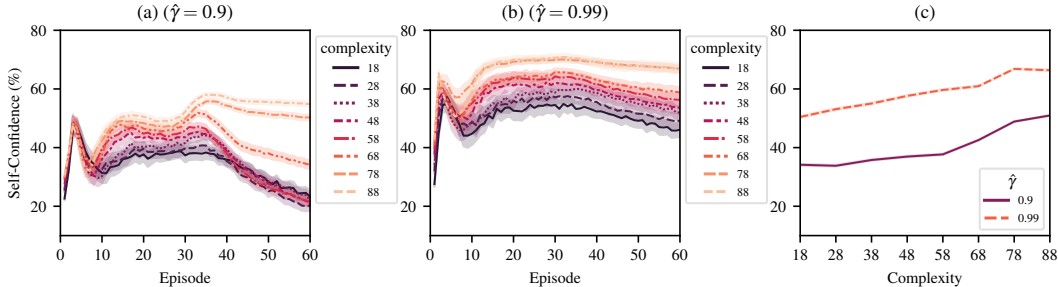

Figure 4: Sensitivity to environment complexity measured with the metric $M2$. 600 agents are trained on 8 mazes of dimension $(10, 10)$ and complexity $(18, 28, 38, 48, 58, 68, 78, 88)$. (a,b) Self-confidence vs. episode with two different discount factors of trust. $\hat{\gamma} = 0.9$ in (a) and $\hat{\gamma} = 0.99$ in (b). (c) Self-confidence tendency per agent vs. complexity in both cases. All curves show an estimate of the central tendency (mean estimator) and 95% confidence interval

We also observe on Figure 4(a) and (b) that the overconfidence peak occurring at the outset of learning is not exclusive. Depending on the trust discount factor and the level of complexity, one or two more peaks appear afterwards. This repetition of overconfidence peaks at the beginning of learning has not been reported in any of the social science research we have studied, and may constitute a new hypothesis.

**Simulation of the Overconfidence Effect.** One of most recent scientific result on the DKE is the measure of the Overconfidence (OC) effect by Sanchez & Dunning (Sanchez & Dunning, 2018). We extracted a dataset of the measure of the self-confidence from their paper and tuned the SelfQ-Trust hyperparameters to reproduce the results of their Study 2. Using the hypermarameter sensitivity study, we have indicated the trend. We then used a brute force approach to find suitable hyperparameters. Parameters are: Maze $(10, 10)$ of complexity $= 88$, 60 agents trained, 60 episodes of learning MDP as trust model, $\alpha = 0.1$, $\gamma = 0.99$, $\hat{\alpha} = 0.05$, $\hat{\gamma} = 0.99$. Results are shown in Figure 5.

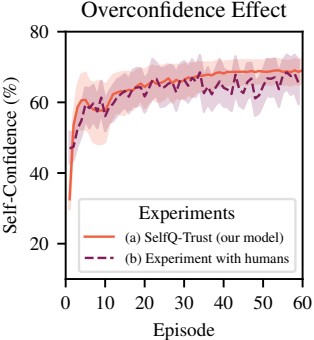

## 6 CONCLUSION

In this paper we provide a new definition of trust for computer science. It is *praxis-oriented* allowing for the development of numerical models. We introduce SelfQ-Trust, a RL algorithm for measuring self-confidence of any human like entity that can be modelled by an intelligent agent trying to solve a MDP. Our method is inspired by the latest findings in neurosciences and RL and shows convincing results, in line with research on the Dunning-Kruger Effect in social psychology. Moreover, we show that numerical simulations of our model allows to formulate new hypotheses; e.g. the intensity of the overconfidence effect depends on the complexity of the task to be learned.

Figure 5: SelfQ-Trust reproduces the results of an experiment involving human volunteers.

Further experiments can be carried out to strengthen the evidence: (1) Justify the use of "distributional reinforcement learning" with an experiment showing that a model with a fixed trust reward do not work. However, we need to study our empirical design in a more theoretical way, notably using the Bellemare operator of distributional reinforcement learning theory. (2) An ablation study to assess the need for dynamic reinforcement. (3) An ablation study on the exploitation of the trust model, with and without time-decaying uncertainty on the prior trust level. (4) Experiments with different base algorithms e.g., DQN and A2C would evaluate the usefulness of SelfQ-Trust in more complex RL formulations, and whether the results hold. (5) Experiments involving human experimenters and serious games to further investigate the overall design hypothesis.

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

## A  ALGORITHM

---

**Algorithm 1: SelfQ-Trust** estimating self-trust of an agent $f$ learning to solve a MDP.

---

**Data:** trust learning rate $\hat{\alpha} \in ]0,1]$, $\hat{\epsilon} = 1$

1   Initialize $\hat{Q}_f(s) \sim \mathcal{N}(\mu, \sigma^2) - 1$ for all $s \in \mathcal{S}$ with e.g. $\mu = 0$ and $\sigma^2 = \frac{card(\hat{Q}_f(s))}{4} = \frac{2n+1}{4}$

2   Initialize intensity of trust reward $\hat{R}_f = 0$

3   Initialize empty list of scores for each past episode $(v_j)_{1 \le j \le i}$

4   **foreach** *episode $i$* **do**

5      $s \leftarrow s_0$; $steps \leftarrow 0$

6      **foreach** *step of episode - until $s$ is $s_T$* **do**

7          Get action $a$ w.r.t. MDP model $f$ and state $s$

8          $n \leftarrow$ uniform random number $\in [0,1]$

9          **if** $n < \hat{\epsilon}$ **then** $\hat{a} \leftarrow 2n$

10         **else**

11             $T \leftarrow$ uniform random number
                     $\in \{\frac{j}{n}\}_{-n \le j \le n} \cap [\max(-1, T_f(f,s) - \hat{\epsilon}), \min(1, T_f(f,s) + \hat{\epsilon})]$

12             Get trust action $\hat{a}$ from $T$: $\hat{a} \leftarrow n(1+T)$

13         Take action $a$ in $env$; learn MDP

14         Learn trust: *LearnTrust($\hat{a}, s, s'$)*

15         $s \leftarrow s'$

16         $steps \leftarrow steps + 1$

17      **if** $i \notin \{0,1\}$ **then** $\hat{R}_f \leftarrow -\dfrac{v_i - \max\limits_{1 \le j < i}(v_j)}{v_i}$

18      Append $-steps$ to $(v_i)$

19      Decay $\hat{\epsilon}$

20   **Function** *LearnTrust($\hat{a}, s, s'$)*

21      $\delta \leftarrow \hat{\alpha} \left[ (\frac{\hat{a}}{n} - 1)\hat{R}_f + \hat{\gamma} \max\limits_{\hat{a}' \in \hat{\mathcal{A}}} \hat{Q}_f(s', \hat{a}') - \hat{Q}_f(s, \hat{a}) \right]$

22      Get $X \in \mathbb{R}^{4n+1} \sim \mathcal{N}(0,1)$; $Y \leftarrow (X_{2n-\hat{a}}, ..., X_{4n-\hat{a}})$

23      $\hat{Q}_f(s) \leftarrow \hat{Q}_f(s) + \delta Y$

---

