# OpenReview forum: "An Attempt to Model Human Trust with Reinforcement Learning"
_ICLR.cc/2022/Conference — ICLR 2022 Submitted_

### Official Review · Reviewer_owda · 2021-11-02

**Correctness:** 1
**Technical Novelty And Significance:** 1
**Empirical Novelty And Significance:** 1
**Recommendation:** 3
**Confidence:** 4

**Main Review:**

The idea of modeling self-confidence is indeed very interesting and fascinating. Having said that, unfortunately the paper is not completely convincing. The authors try to map a series of different concepts from other disciplines into reinforcement learning concepts, but the actual rationale at the basis of the composition of the different resulting component is unclear.

The key formulas in the paper (Formulae 3 and 5-7) are not really justified. They appear as given "modeling assumptions" in a sense. Moreover, the derivation of the trust vector (Formula 3) is not sufficiently discussed in this work. Why can we call it a trust vector? The introduction of distributional learning is also not clearly introduced in the paper. The authors say: "instead of applying learning only on the trust level t prior to the relationship, it is applied on all the possible trust levels". Why is this needed?

The reviewer really struggles to understand how the model of the dopaminergic system (Section 3.3) fits with the rest of the paper. At the end, it seems that this is not actually used in the rest of the paper, except for a potential link with the idea of reinforcement learning, but this "analogy" is not new per se.

The mathematical foundation of the "dynamic learning" presented in Section 4.2 is rather unclear. Its introduction is again not fully justified. Why do you need to introduce it? What does it really add? This should have been evaluated in the paper in my opinion.

In the formulation, the authors refer to the "multi-agent" case, but it is difficult to understand how this would be implemented in practice (also the notation for multiple agents is not completely clear). The reviewer does not understand why you need a different notation for the single agent as well. Should not be the same in a sense?

The authors also do not discuss the selection of the values of the models, which are difficult to set a priori.

The evaluation is based on experiment about self-confidence, including an experiment concerning the selection of the Dunning-Kruger effect. It seems to me that this is a specific case and, in any case, it is difficult to understand the actual effect on the results of the model. Are you really model self-confidence? That is not clear from the mathematical model - for this reason, the interpretation of the results provided by the authors might be questionable.

More detailed comments:

Section 1.1: these are only two possible definitions of trust. It seems to me that this is an oversimplification in a sense.

Section 1.2: it is unclear how this section is related to the rest of the work. These concepts are not really explored in this work.

Section 1.3: the three questions listed here are not really unanswered in the paper in my opinion.

Section 5: this section is essentially about self-trust. The reviewer wonders if it would be better to rewrite the paper with a focus on self-confidence?


**Summary Of The Paper:**

The paper describes an approach to model "self-confidence". The authors use the term trust, but the work focusses mainly on confidence on a certain personal decision. The paper is not about trust in other agents. The model is evaluated using a simulation of a maze where the agent has different degrees of self-confidence. In general, I would say that the contribution is rather limited since the authors use a mathematical model, which is an integration of different components without a clear set of underlying principles.

**Summary Of The Review:**

The problem of trust is interested, but the paper actually focuses on "self-confidence". The assumptions and formulation of the mathematical model are not justified. Several concepts are introduced in the paper, but they are not really used in the model. The findings of the simulations are not insightful, since the model is based on assumptions that are not based on a clear rationale.

---

> ### Author Response · Authors · 2021-11-19
> **Author answers to reviewer questions (3/3)**
>
> **Q3: The mathematical foundation of the "dynamic learning" presented in Section 4.2 is rather unclear. Its introduction is again not fully justified. Why do you need to introduce it? What does it really add? This should have been evaluated in the paper in my opinion.**
>
> See previous answer for a better justification of the technique. Please note that we plan to run an ablation study on the dynamic reinforcement in our future works (see conclusion).
>
> **Q4 In the formulation, the authors refer to the "multi-agent" case, but it is difficult to understand how this would be implemented in practice (also the notation for multiple agents is not completely clear). The reviewer does not understand why you need a different notation for the single agent as well. Should not be the same in a sense?**
>
> We have reordered the paragraphs to separate the presentation of the multi-agent case from the application in the single-agent case.
>
> The notation is the same in both cases. For the single-agent case, it is only necessary to replace $g$ by $f$ (in that case, the trustor and the trustee are the same agent). The code repository https://github.com/selfQtrust/code provides in fact an implementation of the multi-agent case. However, (1) some additional parts are needed in this case, in particular a compromise design between the opinions of several agents (see the def __choose_action_exploit function, line 197 of https://github.com/selfQtrust/code/blob/master/framework/qlearning.py). We are not quite satisfied with this design and would prefer to take it, for example, by using a contextual bandit system and/or an adversarial method (see also answer to reviewer GxNj's Q3). (2) New experiments are needed for this case, and will be the subject of future work.
>
> **Q5 The authors also do not discuss the selection of the values of the models, which are difficult to set a priori.**
>
> You refer to the last experiment. Using the hyperparameter sensitivity study, we have indicated the trend. We used a brute force approach, but we are planning to deal with this in more detail in our future experiments. See the revised version of the "Simulation of the Overconfidence Effect" section of the experiments.
>
> **Q6 The evaluation is based on experiment about self-confidence, including an experiment concerning the selection of the Dunning-Kruger effect. It seems to me that this is a specific case and, in any case, it is difficult to understand the actual effect on the results of the model. Are you really model self-confidence? That is not clear from the mathematical model - for this reason, the interpretation of the results provided by the authors might be questionable.**
>
> On the assessment of the model, you're right. In our research methodology, we have defined a set of experiments to assess the model. More experiments are planned, but, as we found it remarkable, we decided to publish the reproduction of the DKE.
>
> For the mathematical model, please consult the modifications done following your first two questions.
>
> **Section 1.1: these are only two possible definitions of trust. It seems to me that this is an oversimplification in a sense.**
>
> We understand your point, but Luhmann's definition is widely accepted in the community, and we chose to keep it.
>
> **Section 1.2: it is unclear how this section is related to the rest of the work. These concepts are not really explored in this work.**
>
> This remark is shared with the other reviewers, and we understand that this part is not really relevant in the document. Section 1.2 is an introduction for our research in all, and not only for this contribution. In the new version of the paper, we remove section 1.2 and focus on self-confidence. We give the overview of the research with answer provided to Q1 of reviewer dEMd.
>
> **Section 1.3: the three questions listed here are not really unanswered in the paper in my opinion.**
>
> We fully agree. We have removed the research questions in the new version of the paper to focus on hypotheses for which we provide evidence through our experiments.
>
> **Section 5: this section is essentially about self-trust. The reviewer wonders if it would be better to rewrite the paper with a focus on self-confidence?**
>
> Thank you for this remark. The new version of the paper is written along this advice.

---

> ### Author Response · Authors · 2021-11-19
> **Author answers to reviewer questions (2/3)**
>
> **Q2: The reviewer really struggles to understand how the model of the dopaminergic system (Section 3.3) fits with the rest of the paper. At the end, it seems that this is not actually used in the rest of the paper, except for a potential link with the idea of reinforcement learning, but this "analogy" is not new per se.**
>
> The model of the dopaminergic system is used in the "Dynamic reinforcement" and "Distributional reinforcement learning" part of the "4.2 Learning Principles" section, but the link is not clearly explained.
>
> Firstly, we hypothesized that trust is a process that is built up in the human brain. Secondly, we define this process as a bet - therefore fundamentally reward-based. We therefore require that our model takes into account the latest neuroscientific knowledge about the reward circuit in the human brain.
>
> The reward prediction error (RPE) hypothesis of dopamine — the discrepancy between observed and expected reward — has become central to research at the intersection of neuroscience and computer science because RPE is precisely the signal that a RL system would need to update reward expectations \citep{montague1996framework, schultz1997neural}. Dabney et al. (2020) findings provide strong evidence to support the hypothesis that the brain represents future rewards not as a single scalar quantity, but rather as a probability distribution. The scientific breakthrough is to consider the RPE hypothesis as representing multiple future rewards simultaneously and in parallel. Specifically, one can look at Figures 5.a and 5.b of their paper, which compare the decoding of reward distributions from neural responses with classical TD decoding and distributional TD decoding - showing that distributional reinforcement learning seems to be a necessary technique to introduce if we want to model the brain's reward circuit as closely as possible to the latest scientific knowledge.
>
> We apply the hypothesis to our model of trust learning. We therefore require that the bet made by the trustor in a trust relationship with a trustee must give rise to several simultaneous rewards over all possible trust levels (see "Distributional reinforcement learning" section) and that the reward magnitude must be updated according to an observation of the result of the bet (see "Dynamic reinforcement" section).
>
> Please see the new section "Using the neural realization of distributional reinforcement learning hypothesis." of the section, "4. Method" in the revised version of the paper.
>
> @article{montague1996framework,
>   title={A framework for mesencephalic dopamine systems based on predictive Hebbian learning},
>   author={Montague, P Read and Dayan, Peter and Sejnowski, Terrence J},
>   journal={Journal of neuroscience},
>   volume={16},
>   number={5},
>   pages={1936--1947},
>   year={1996},
>   publisher={Soc Neuroscience}
> }
>
> @article{schultz1997neural,
>   title={A neural substrate of prediction and reward},
>   author={Schultz, Wolfram and Dayan, Peter and Montague, P Read},
>   journal={Science},
>   volume={275},
>   number={5306},
>   pages={1593--1599},
>   year={1997},
>   publisher={American Association for the Advancement of Science}
> }

---

> ### Author Response · Authors · 2021-11-19
> **Author answers to reviewer questions (1/3)**
>
> Dear reviewer,
>
> Thank you for taking the time to read our paper and for your detailed questions on the technical part which helped us to improve the formulation of the mathematical model.
>
> In the new version of the paper, we have shortened the introduction to stay focused on the rest of the paper. We have elaborated on section "4. Method" to make the presentation of our contribution more detailed and precise. Specifically, we introduced a new section "Using the neural realization of distributional reinforcement learning hypothesis" to justify the introduction of distributed reinforcement learning and to link our model to recent findings in neuroscience about the functioning of reinforcement circuits and model of the dopaminergic system. We rewrote section 4 in an effort to better justify formulae 3 and 5-7. We have reordered the paragraphs to separate the presentation of the multi-agent case from the application in the single-agent case. As you suggest we recognize that additional experiments are needed to provide further evidence of the relevance of the approach, and better tuning of the setup. They are listed in our future work.
>
> **Q1 : The key formulas in the paper (Formulae 3 and 5-7) are not really justified. They appear as given "modeling assumptions" in a sense. Moreover, the derivation of the trust vector (Formula 3) is not sufficiently discussed in this work. Why can we call it a trust vector? The introduction of distributional learning is also not clearly introduced in the paper. The authors say: "instead of applying learning only on the trust level t prior to the relationship, it is applied on all the possible trust levels". Why is this needed?**
>
> A1: Formula (3) is the application of Q-learning in the state-trust actions space. The space of trust actions is discretised with $card(\hat{A}) = 2n$. Applying the Q-learning principle, the trust level estimated by the model is therefore the one obtained thanks to the index of the trust action for the max of the Q-value. To report it in the interval $[-1,1]$, we divide by $n$ and subtract $1$. The notion of "Q-value trust vector" refers to a Q-value estimated for all possible trust levels (i.e. (k/n)(k ín [[-n,n]]) knowing a state and an action.
>
> Formula (5) is a heuristic that weights the reward of the trust model according to the score that the agent measures at the end of a learning episode of the underlying MDP. The idea is as follows: we consider that the agent stores its scores in a $memory$ list of fixed-size. if the agent beats the highest score it has stored in the last $card(memory)$ episodes, one can consider that it was right to trust itself over the past episode, driving to a positive reward for the next episode. Conversely, if the agent does less well than its highest score, one can consider that it was wrong to trust itself on the past episode, driving to a negative reward for the next episode. The magnitude of the reward will be proportional to the difference with the highest score observed in the memory. We consider to run further experiments on the choice of this heuristic to better justify the model. Please note that in the revised version of the paper, Formula (5) is now Formula (6) because we have reordered the paragraphs to separate the presentation of the multi-agent case from the application in the single-agent case.
>
> Formula (6) is the application of Q-value updating method by applying a TD method. However, we do not apply equation (2) to the Q-value of the a priori trust level, but to all possible Q-values of trust actions at the given state-action pair with a realisation of a normal distribution centered on the a priori trust level. This choice is motivated by the empirical observation that the learned trust level converged very quickly to $+1$ or $-1$ when we simply applied formula (2) to the Q-value of the a priori trust level. Doing so, we add Gaussian noise to the reinforcement, which makes trust learning efficient.
>
> Obviously, we should have justified this choice with an additional experiment that showed that a model with a fixed trust reward did not work. However, we need to study our empirical design in a more theoretical way, notably with the contributions of distributional TD learning and the Bellemare operator. Please note that in the revised version of the paper, Formula (6) is now Formula (5).
>
> Formula (7) enables involving an uncertainty on the level of trust that the trustor grants to the trustee in case of exploitation of the trust model. The uncertainty is decreasing with time, i.e. it is dependending on the $\hat{\epsilon}$ which refers to the probability of choosing to explore the trust model. We consider to run an ablation experiment to motivate ther introduction of the uncertainty a posteriori.
>
> Your questions help us elaborated on section "4. Method" to make the presentation of our contribution more detailed and precise. Please see the new version of the paper with the revised section, "4. Method"

---

### Official Review · Reviewer_fiz1 · 2021-11-02

**Correctness:** 2
**Technical Novelty And Significance:** 3
**Empirical Novelty And Significance:** 2
**Recommendation:** 3
**Confidence:** 3

**Main Review:**

The paper tackles a novel trust problem with several (economics, psychological and engineering) implications for designing agents with appropriate metacognition for decision-making. However, there were a few conceptual concerns that need to be addressed for me:

Main comments:
- There are conceptual gaps between the definition of trust introduced in Section 1 ‘an assessment of risk by an individual who makes a rational choice’, and Section 4 ‘is both a decision-making process in a relationship between two intelligent agents and the measure by which the decision is made’. Particularly regarding the measure by which the decision is made – this speaks to the metacognition and not an assessment of risk. It would be helpful to explain exactly how trust differs from meta-cognition. Additionally, to avoid confusing the reader to introduce the definition right at the beginning.
- The introduction was too long winded and not enough description about the research question e.g., what is a digital model? What are the evolution drivers? The paper speaks to it briefly in Section 3.3 regarding genetic fitness – is this the complexity? However, links are tenuous at best given that it jumps straight to learning in the brain, and not evolution. This would be interesting point to pursue, without jumping to dopamine, and reward circuitry. Additionally, the related literature needs to focus on economics, neuroeconomics, and social psychology literature e.g., Mazor, M., Fleming, S.M. The Dunning-Kruger effect revisited. Nat Hum Behav 5, 677–678 (2021). , Cox, J.C. (2004) How to Identify Trust and Reciprocity. Games and Economic Behavior, 46: 260-281.
- The paper is premised on Berg’s hypothesis. Therefore, it was quite surprising to see that the simulations used a grid-world instead a more appropriate algorithm where the agent had to make a pay-off decision. This would allow the simulation results to be evaluated in comparison to the  ‘real-life world’ scenarios that were mentioned in the abstract. Additionally, it would be interesting to evaluate the DK effect in more realistic scenarios using this Q-value trust function. Additionally, it would be interesting, and useful, to evaluate under what conditions this isn’t replicated. Currently, this effect is barely visible when the environment set-up is simple (6x6).
- The paper mentions that this trust function can be added to other model-free RL algorithms. This claim should be supported by additional experiment with different base algorithms e.g., DQN and A2C. This would help the reader evaluate the usefulness of this in more complex RL formulations, and whether the results hold.
- In the introduction, there was a discussion around the different types of actions available e.g., assured and chosen. The simulations don’t prescribe this separation in action type or ability to control the environment. Perhaps, as mentioned in the introduction this might be relevant for multi-agent systems. However, I would like to understand how the DKE, or the learning differs depending on the type of actions an agent makes.

Minor comments:
- Graphical representation of the algorithm, and exactly how the trust learning function can be included in other model-free RL algorithms. It is not spelt out clearly exactly how the trust learning function would benefit the other algorithms, and what the additional complexity costs associated with this are.
- Please review and update typing errors and grammar throughout.
- Please explain all variable / parameters being introduced.
- There is repetition in the paper, e.g., some quotations are introduced twice, that should be removed.


**Summary Of The Paper:**

The paper presents a computational formulation of trust. For this, it introduces a Q-learning trust algorithm; where the agent's trust levels (at self and when dealing with another agent) can be evaluated using a Q-value function. This is achieved by: i) varying the trust reward received, ii) introducing a distributional TD learning over the possible trust levels, and iii) using an epsilon greedy strategy to learn the Q-value function for the trust model. The framework is face-validated in a grid-world where the agent must navigate nxn maze to reach a particular end location.

Interestingly, the paper proposes that this trust learning function proposed can be introduced in other model-free RL algorithms.

**Summary Of The Review:**

The paper nicely presents a simple function for defining trust. However, the conceptual and experimental shortcomings need to be addressed. Specifically, appropriate experiments need to be run to evaluate the observed effects and their implications on understanding human behaviour. Ideally, the paper would simulate previously evaluated human experiments and compare those results with their simulation analysis. The metrics used would also need to be comparable.

---

> ### Author Response · Authors · 2021-11-19
> **Author answers to reviewer questions (2/2)**
>
> **Q6: In the introduction, there was a discussion around the different types of actions available, e.g., assured and chosen. The simulations don’t prescribe this separation in action type or ability to control the environment. Perhaps, as mentioned in the introduction, this might be relevant for multi-agent systems. However, I would like to understand how the DKE, or the learning, differs depending on the type of actions an agent makes.**
>
> In the revised version of the paper, we have removed the section that explains the distinction between chosen and decided actions, which refers to Luhmann's theory. However, your comment will be quite useful when we will scale up the model - whether Hundayi's hypothesis holds meaning that and all actions related to a trust relationship are functionally equivalent - or whether we should introduce the distinction between chosen and assured actions, as Luhmann does.

---

> > ### Comment · Reviewer_fiz1 · 2021-11-28
> > **Paper reviews**
> >
> > Thank you for carefully updating the paper, I appreciate your efforts and believe the idea has merits.
> >
> > However, the execution and experiments need further work despite the revisions. Specifically, I would encourage the others to better organise the introduction and have more appropriate experiments to evaluate their quantitative formulation. I would be interested in reading such a paper that: a) uses more trust-relevant paradigms beyond the grid-world task and b) validate the applicability of the algorithm for other model-free RL algorithms.

---

> ### Author Response · Authors · 2021-11-19
> **Author answers to reviewer questions (1/2)**
>
> Dear reviewer,
>
> We sincerely thank your summary of our contribution and your relevant suggestions.
>
> To be more in line with the actual contribution of the paper, we have much simplified introduction to keep focus on the definition of trust we simulate in our work. This led us to suppress section 1.2. We clarified the definition of trust, and the drivers used (brain, not evolution). In section 4 we have given more details on the formulas/parameters. Finally, we provide in conclusion some insights on the future work, especially additional experiments that are needed to provide further evidence of the relevance of the approach.
>
> **Q1: The introduction was too long-winded and not enough description about the research question, e.g., what is a digital model? What are the evolution drivers? The paper speaks to it briefly in Section 3.3 regarding genetic fitness – is this the complexity? However, links are tenuous at best given that it jumps straight to learning in the brain, and not evolution. This would be interesting point to pursue, without jumping to dopamine, and reward circuitry.**
>
> As your question echoes those of reviewer owda (see Q8) and GxNj (see Q1), we consider that the section 1.2 is out of scope and suppress it. See answer to GxNj's Q1 for a more complete answer. We recognize that the links to dopamine and reward circuitry research were not clearly enough justified in the first version of the paper. We have therefore introduced a new section entitled "Using the neural realization of distributional reinforcement learning hypothesis." in section "4. Method" to show that it is directly learning in the brain and not evolution.
>
> **Q2 Additionally, the related literature needs to focus on economics, neuroeconomics, and social psychology literature e.g., Mazor, M., Fleming, S.M. The Dunning-Kruger effect revisited. Nat Hum Behav 5, 677–678 (2021). , Cox, J.C. (2004) How to Identify Trust and Reciprocity. Games and Economic Behavior, 46: 260-281.**
>
> Thanks for the related literature which we add in related work section. We also believe that our research could benefit to important questions raised in macroeconomics; this is why we have added the 3rd paragraph of the section "2. Related Works" in the revised version of the paper (see paragraph beginning by "In economics,").
>
> **Q3 Graphical representation of the algorithm, and exactly how the trust learning function can be included in other model-free RL algorithms. It is not spelled out clearly exactly how the trust learning function would benefit the other algorithms, and what the additional complexity costs associated with this are.**
>
> Thanks for this very valuable question. We have added a graphical representation of the algorithm focussing on the distributional reinforcement learning in section "4. Method" to make the presentation of our contribution more detailed and precise.
> We delete the claim that the trust learning function would benefit other algorithms and add it as future work in the conclusion - as we need more time to finish our experiments on this point.
>
> **Q4 The paper is premised on Berg’s hypothesis. Therefore, it was quite surprising to see that the simulations used a grid-world instead a more appropriate algorithm where the agent had to make a pay-off decision. This would allow the simulation results to be evaluated in comparison to the ‘real-life world’ scenarios that were mentioned in the abstract. Additionally, it would be interesting to evaluate the DK effect in more realistic scenarios using this Q-value trust function. In addition, it would be interesting, and useful, to evaluate under what conditions this isn’t replicated. Currently, this effect is barely visible when the environment set-up is simple (6x6).**
>
> You highlight an important point that is planned in our research. Specifically, one of our priorities in the coming months is to develop serious games to be able to compare the results of our algorithm with our own collected dataset - all other things being equal. In this context, Berg's hypothesis will also be used to measure the a priori confidence level of each individual in the cohort, as is done in the literature. See the last point on the revised version of the paper.
>
> **Q5: The paper mentions that this trust function can be added to other model-free RL algorithms. This claim should be supported by additional experiment with different base algorithms, e.g., DQN and A2C. This would help the reader evaluate the usefulness of this in more complex RL formulations, and whether the results hold.**
>
> As we did not demonstrate this with an experiment in this paper, we have removed the claim from part "4. Method". We have mentioned the point in the future work in the conclusion of the revised version of the paper.

---

### Official Review · Reviewer_GxNj · 2021-11-03

**Correctness:** 2
**Technical Novelty And Significance:** 2
**Empirical Novelty And Significance:** 1
**Recommendation:** 3
**Confidence:** 3

**Main Review:**

I think understanding trust is of obvious importance, so from that perspective the work is well-motivated. However, I think the paper as it currently stands spends too much time on motivation and setup and not nearly enough either explaining its specific contribution in detail, or providing a strong payoff to all the setup.

Specifically, the paper spends nearly a page and a half on a near-philosophical digression on trust and confidence -- what is the relevance / importance of this to the contribution? Similarly with the high-level claims on the importance of trust to society -- it's fairly obvious that understanding trust is important, and an extended discussion of this is not needed. The paper spends another half-page reviewing connections from RL to the modeling the dopaminergic system -- this connection is worth mentioning and citing but multiple paragraphs of discussion seem out of place). It spends a fair bit of space motivating the modeling of trust in a population of agents, even though the actual contribution is only a self-trust model. Additional space is spent on using a scrum master and agile development as a worked example -- I'm not sure how familiar this would be to the ICLR audience. At the same time, the specific algorithm description is left to the appendix, and from reading it I'm still not sure how the trust-learning algorithm interfaces with the regular RL algorithm? It seems like the trust algorithm has its own action space and reward space -- is it a sort of off-policy contextual bandit where the regular RL algorithm manages state transitions and SelfQTrust learns to choose trust actions given a state it doesn't get to control? I'm still not sure what exactly was done and confused about the distinction between what the general multi-agent framework is here, vs what was actually implemented and evaluated.

In addition, I have some technical puzzles and concerns:
- If T_x() leads to one of two outcomes (i.e. it is a Bernoulli random variable) why the parameterization of p(act) - p(not_act)? Wouldn't p(act)+p(not_act)=1, implying that T_x(...) could just be p(act) w.l.o.g.?
- Why does the agent's choice to perform an action *occur* with a probability that's a function of s'? Considering s' is not known, transition probability should be defined for each action given state s'. T(s, a, s') seems like the action-conditional transition probability, i.e. a property of the environment.
- Expr 1 and 2 notation is underspecified: what is the expectation over? I believe it is the next action given some policy. More importantly, if the tickmark `'` means "next" (action/state), then shouldn't the notation have R' and max Q(s', a)? I.e. next reward and Q value of current action next state.

Finally, there are some typos, including:
- "choosen action" -> chosen.
- "agent choose action" -> chooses.
- "whith the probability".

**Summary Of The Paper:**

The submission proposes a model of the dynamics of trust from a reinforcement learning perspective. It provides a simulation of the Dunning-Kruger effect in a simple grid world using this model.


**Summary Of The Review:**

I am not an expert in trust modeling, but even so I think the paper as framed is mistargeted for the conference. It needs to have much shorter / pithier motivation and review, and much more detailed and precise explanation of what was actually done. In addition, there need to be some sort of payoff (i.e. new results / insights) from all this framing -- as-is, there is the DK and OC effects, but to the extent there's a prediction of new effects (e.g. the location of the first confidence peak) the paper dismisses it as a potential artifact of the setup, so I'm not sure what to make of it.

---

> ### Author Response · Authors · 2021-11-19
> **Author answers to reviewer questions (2/2)**
>
> **Q4: If T_x() leads to one of two outcomes (i.e. it is a Bernoulli random variable) why the parameterization of p(act) - p(not_act)? Wouldn't p(act)+p(not_act)=1, implying that T_x(...) could just be p(act) w.l.o.g.? Why does the agent's choice to perform an action occur with a probability that's a function of s'? Considering s' is not known, transition probability should be defined for each action given state s'. T(s, a, s') seems like the action-conditional transition probability, i.e. a property of the environment.**
>
> Thank you for this important question which allows us to clarify our method. We believe that there is a problem of understanding due to a wrong formulation of definition 1 on our side. The sentence "The trust level $T_f(g, s) in [−1, 1]$ is calculated from the subjective measure of the probability that the bet will come true", inherited from Marsh's formalism, is indeed not accurate in our formal context - in the sense that it introduces a notion of probability that does not apply to our model. In his recent work - which we had not yet fully explored at the time of writing our manuscript - the philosopher Mark Hunyadi introduces a new definition of trust that seems to us to be much more appropriate to our method: "Trust is a bet on the behavioural expectations of things (e.g. that the ground will support me when I walk), people (that the driver of the car I pass will obey the rules of the road), and institutions (that the money I use for my transactions will have some value) (translated from French by us ; read e.g. https://journals.openedition.org/lectures/47235 for a quick overview of Hunyadi's theory). Defining trust, in its functional aspect, by means of a bet on the future behaviour of another agent with whom one is in a relationship leads naturally to the use of game theory or reinforcement learning. Keeping in mind that trust is also a measure of the a priori that the trustor grants to the trustee - and that this measure will evolve with learning - leads to the choice of reinforcement learning. In conclusion, we would like to propose this new definition to clarify our point:
>
> Definition 1 (Trust): Trust is a bet made by an intelligent agent on the behavioural expectations it has of another agent or of itself. Considering that an agent $f$, named trustor, trusts another agent $g$, named trustee, we define a level of trust $T_f(g) in [-1, 1]$ as a measure of the subjective certainty of the trustor $f$ that its bet will come true if the level is positive, and false if the level is negative.
>
> And, because experiments are forcussing on self-confidence, we add also a definition of self-confidence, which is the specialisation of definition 1 when the trustor and the trustee are the same agent:
>
> Definition 2 (Self-confidence): Self-confidence is a bet made by an intelligent agent on its future action. We define a level of self-confidence $T_f \\in [-1, 1]$ as a measure of the subjective certainty of the agent $f$ that future action it will take is optimal with respect to the task it is currently solving if the level is positive, and non optimal if the level is negative.
>
> Please see the new version of the paper with the revised section "4. Methods" including these new definitions.
>
> **Q5: Expr 1 and 2 notation is underspecified: what is the expectation over? I believe it is the next action given some policy. More importantly, if the tickmark ' means "next" (action/state), then shouldn't the notation have R' and max Q(s', a)? I.e. next reward and Q value of current action next state.**
>
> Right. We correct the notations using equations (3.20) and (6.8) of the reference book \\cite{sutton2018reinforcement}.
>
> @book{sutton2018reinforcement,
>   title={Reinforcement learning: An introduction},
>   author={Sutton, Richard S and Barto, Andrew G},
>   year={2018},
>   publisher={MIT press}
> }

---

> ### Author Response · Authors · 2021-11-19
> **Author answers to reviewer questions (1/2)**
>
> Dear Reviewer,
>
> Thanks you for taking the time to read our paper and provide us valuable feedback.
>
> In the new version of the paper, we have shortened the introduction and we have endeavoured to clearly address the global perspective. We have shown the importance of reviewing the definitions of trust used so far in computer science. This justifies our choice to start from new hypotheses, in particular those resulting from recent and related research in neuroscience and reinforcement learning. We have elaborated on section "4. Method" to make the presentation of our contribution more detailed and precise. Specifically, we have updated the central definition of trust and corrected what we now consider to be an error inherited from the literature on trust modelling consisting in the use of frequentist probability theory. We recognize that additional experiments are needed to provide a strong payoff to all the setup ; they are listed in our future work. On the other hand, we hope to show the importance of new approaches like ours with the new version of the paper - as we need the assessment of the community to go further on.
>
> **Q1: Specifically, the paper spends nearly a page and a half on a near-philosophical digression on trust and confidence -- what is the relevance / importance of this to the contribution?**
>
> As your question echoes those of rewiewer owda (see Q8) and fiz1 (see Q1), we consider that the section 1.2 is out of scope and supress it. Yet our motivation writing this section were as follows. Many scientific papers in computer science lack a precise and operational definition of trust, making the results biased by the implicit understanding of the concept of their authors. This fact is widely acknowledged, as many papers of which we are aware point to the "lack of definition of trust". Since trust is a structure of a social system, we believe that we can overcome this difficulty by drawing on the scientific knowledge base in sociology. But to make it a practical and useful framework for computer researchers, we have tried to present their theory - in particular Luhmann's - in a didactic way with examples that are meaningful for computer engineer / scientists - the choice of agile methods as a common thread.
>
> Please see the new version of the paper with the revised introduction.
>
> **Q2: the specific algorithm description is left to the appendix, and from reading it I'm still not sure how the trust-learning algorithm interfaces with the regular RL algorithm?**
>
> It is probably easier to answer your question by looking at our implementation, available in the github repo pointed to in the paper (https://github.com/selfQtrust/code)
>
> If you look at the framework/qlearning.py file (https://github.com/selfQtrust/code/blob/master/framework/qlearning.py), you will see that the QTrust class (lines 111 and following) - implementing Algortihm 1 left to the appendix - inherits from the QLearning class which implements a simplest Q-Learning algorithm. Thus, by keeping the same interfaces, it is technically quite easy to replace Q-Learning with more sophisticated algorithms.
>
> **Q3: It seems like the trust algorithm has its own action space and reward space -- is it a sort of off-policy contextual bandit where the regular RL algorithm manages state transitions and SelfQTrust learns to choose trust actions given a state it doesn't get to control? I'm still not sure what exactly was done and confused about the distinction between what the general multi-agent framework is here, vs what was actually implemented and evaluated.**
>
> The trust algorithm has its own action space and reward space: yes. is it a sort of off-policy contextual bandit where the regular RL algorithm manages state transitions and SelfQTrust learns to choose trust actions given a state it doesn't get to control? Yes, that seems a good way to sum up our idea. Moreover, thank you for the question which sounds with a pending work for the implementation of the multi-agent version. The Epoch-Greedy Algorithm for Contextual Multi-armed Bandits (see Figure 1: Exploration by epsilon-greedy in epochs in \cite{langford2007epoch}) seems to be a good idea for a better compromise design in the multi-agent case (see def __choose_action_exploit, line 197 of https://github.com/selfQtrust/code/blob/master/framework/qlearning.py)
>
> ref : @article{langford2007epoch,
>   title={The epoch-greedy algorithm for contextual multi-armed bandits},
>   author={Langford, John and Zhang, Tong},
>   journal={Advances in neural information processing systems},
>   volume={20},
>   number={1},
>   pages={96--1},
>   year={2007},
>   publisher={Curran Associates Red Hook, NY}
> }

---

### Official Review · Reviewer_dEMd · 2021-11-06

**Correctness:** 3
**Technical Novelty And Significance:** 3
**Empirical Novelty And Significance:** 3
**Recommendation:** 6
**Confidence:** 3

**Main Review:**

Pros:
1.	The narrative perspective is interesting and the analogy from trustor-trustee advisory is intuitive, and easy to relate to the technical formulations.
2.	More contemporary concept like distributional reinforcement learning is employed for learning multiple trust levels, and the authors proposed a novel dynamic trust reward based on the change in trust level during the learning process.
3.	This paper provides comprehensive experiments on evaluating hyperparameter sensitivity as well as simulating the DKE overconfidence effect and reproduce the results from other works, adding confidence to the experiment findings.

Cons:
1.	Although it is clear that the goal is to construct a trust model in multi-agent scenario, but the ultimate motivation in the bigger picture perspective is not addressed clearly. For example, what is the benefit or application of constructing a trust model?
2.	From a policy performance standpoint, there is no comparison between the performance when the trust model is present/absent (ablation study). Understand that this may not be the primary goal of this paper (also somehow related to the first question), but this would add to the completeness and perhaps spark future interests among readers.

Questions for clarification:
1.	In the background and introduction section, the definitions and terminologies of trust, confidence and available actions are explained as an integral part of the social systems structure. What role does the “confidence” play in the proposed modelling of trust-decision making process? It seems the proposed method is based primarily on “trust” and “confidence” is an
2.	In section 1.3, “… shared narratives tend to become more liquid. As a consequence, confidence is decreasing to the benefit of trust.” This sounds interesting, but is not exactly clear. Can you elaborate more on this? High self-trust level seems to imply higher confidence in your discussion, but why does this statement observe an inverse relationship between the two?
3.	Is there an explanation or hypothesis on why overconfident is more pronounce in higher complexity problems (especially when the paper has a heavy psychological-inspired tone)? I feel it is better to include at least a line of hypothetical explanation rather than leaving it completely as an open question.

Some typos:
1.	Section 1.2 paragraph 3 last sentence: condidence -> confidence
2.	Section 3.1 paragraph 2 second sentence: [-1, 1[ -> [-1, 1]



**Summary Of The Paper:**


The paper proposed an algorithm to model trust numerically in parallel with the policy learning process in multi-agent scenario. The algorithm resembles an overlay on any reinforcement learning algorithm with modification using a dynamic trust reward, depending on trust levels. The proposal is inspired by dopaminergic system and reaffirms the Dunning-Kruger effect (DKE) in societal studies.


**Summary Of The Review:**

The paper takes on an interesting perspective in trust modelling while connecting the biological, humanities, psychological and societal studies inspirations to the proposed method. The technical formulations are easy to follow and  sound.

---

> ### Author Response · Authors · 2021-11-19
> **Author answers to reviewer questions (3/3)**
>
> **Q3: In section 1.3, “… shared narratives tend to become more liquid. As a consequence, confidence is decreasing to the benefit of trust.” This sounds interesting, but is not exactly clear. Can you elaborate more on this? High self-trust level seems to imply higher confidence in your discussion, but why does this statement observe an inverse relationship between the two?**
>
> We have deleted section 1.3 in the new introduction. However, thank you for the question which clarifies a point that will be useful in our future work (especially when we are able to scale up to the multi-agent version). Here are some answers.
>
> In the thinking of the philosopher Bauman, modernity has been accompanied by a shift from a 'solid' to a 'liquid' society. In short, we have moved in a century from a world where social relationships were mostly very stable over the life of an individual (caricaturally, the individual centred on his or her neighbourhood who does not change jobs throughout his or her life) to a world where our social networks have become very dynamic (especially if we are interested in our social projection in the digital sphere).We observe that the common narratives of the 19th century (e.g. money, capitalism, brands) probably had a much longer lifespan than those of the digital world (e.g. yahoo, facebook, blockchain as examples of totems of the digital era)
>
> We expect to answer the second question by conducting experiments at scale the with the multi-agent version of our algorithm (see answer A2dEMd to your previous question), which was not in the scope of this paper.
>
> **Q4: Is there an explanation or hypothesis on why overconfident is more pronounce in higher complexity problems (especially when the paper has a heavy psychological-inspired tone)? I feel it is better to include at least a line of hypothetical explanation rather than leaving it completely as an open question.**
>
> Thank you for this suggestion. We added this line in the paper: "It may be hypothesised that, the more complicated the problem is to solve, the more difficult it will be to learn self-confidence and the more biases in that learning - such as overconfidence - will manifest themselves.". To answer fully, we could not test this hypothesis because the existing dataset we used does not provide the complexity perceived by the agents in the serious games used. This is also why we plan to carry out our own experiments involving humans in our future work.

---

> ### Author Response · Authors · 2021-11-19
> **Author answers to reviewer questions (2/3)**
>
> **Q2: What role does the “confidence” play in the proposed modelling of trust-decision making process? It seems the proposed method is based primarily on “trust” and “confidence” is an**
>
> Your question was truncated, but we think we got it. You're right if you meant that "the proposed method is based primarily on “trust” and “confidence” is a a concept that is not used in the model". For this paper, we do not need to focuss on Luhmann's confidence except for perspectives. However, we have provided a full answer below. Part of the answer has helped us to rewrite our introduction.
>
> "Trust" and "confidence" are defined in Luhmann's work. We have used this theory because it is the most enlightening in our opinion to explain the problems of trust in a social system, in particular at a time of digitalization of human relationship, which is the central problem of our research. Moreover, to the best of our knowledge, most of the literature in sociology makes Luhmann's thought a reference in the field. Nevertheless, the theory remains difficult to access for non-specialists, especially for semantic, linguistic and cultural reasons. This is why we have sought to make a didactic effort in our paper to make it accessible, firstly to our team and our stakeholders.
>
> In short, some elements to illustrate this complexity while answering your question: Luhmann's seminal publication, "Vertrauren", is written in German (1968). In this work, we find the concepts - translated into English - of "interpersonal trust" and "system trust", which are still used in some contemporary publications. However, Luhmann's latest work, in particular his paper "Familiarity, confidence, trust: Problems and alternatives", clearly shows an evolution towards a 2-layer theory: first "familiarity" is a basic and permanent foundation without which social relations would not be possible. Second, "trust" which can be compared to the "interpersonal trust" of his early writings explains the cohesion of a social system based on direct interactions between individuals, while "confidence" - which would be, in our understanding, the "system trust" of his early writings minus "familiarity" - would explain the cohesion of a large-scale social system.
>
> In more recent research, in particular that of the philosopher Hunyadi (in French: "Au début est la confiance" (2020)), this theory of trust is considered to be discontinuous because the intrinsic mechanisms of trust are not the same depending on the scale at which they are observed. Hunyadi proposes a new holistic and continuous definition of trust: "Trust is a bet on the behavioural expectations of things (e.g. that the ground will support me when I walk), people (that the driver of the car I pass will obey the rules of the road), and institutions (that the money I use for transactions will have some value). This seemed to us both to include Luhmann's definition of trust (e.g. "interpersonal trust") and to be practical enough (e.g. the notion of a bet leads quite naturally to defining a reward in the future) to be simulated with reinforcement learning. But we did not want to introduce Hunyadi's thinking at this stage, at the risk of not being understood by social scientists. Thus, it seemed interesting to us to start from Luhmann's definition of "trust" to see if the numerical models, when we will be able to carry out experiments on a large scale (many agents in relatively complex environments), would allow the emergence of other types of trust, or even to validate the continuum of trust hypothesis proposed by Hunyadi.

---

> ### Author Response · Authors · 2021-11-19
> **Author answers to reviewer questions (1/3)**
>
> Dear Reviewer,
>
> Thank you very much for your insightful and helpful comments and suggestions. We are glad you appreciate our work as a useful contribution to the community.
>
> We provide detailed answers as a revised version of the paper to the first Cons in our answer to question labbeled Q1. We will take into account the second Cons when our work on the multi-agent version of the algorithm is complete. We also thank you for your questions for clarification (labbeled Q2 to Q4) as they helps us to (1) shorten the introduction to keep focus on the main contributions and long term vision (Q2) (2) to better envision the future of our research, especially when we are able to scale-up the model (Q3) (3) to improve the discussion of our experimental results as you suggest (Q4).
>
> **Q1: Although it is clear that the goal is to construct a trust model in multi-agent scenario, but the ultimate motivation in the bigger picture perspective is not addressed clearly. For example, what is the benefit or application of constructing a trust model?**
>
> From a practical point of view - e.g. in an industrial context - we observed that there is a lack of tools to forecast the evolution of trust among stakeholders (customers, suppliers, partners) according to the strategic choices of digital transformation. New job positions, such as that of chief trust officer, could be interested in this type of tool. We believe that our research could be a starting point for such future applications.
>
> From a theoretical point of view, we would like to see if the forms of trust that allow us to understand the cohesion of a social system at scale (e.g. Luhmann's "confidence" or "system trust") would not be a simple emergence of interpersonal trust when scaling up. This theoretical contribution would make it possible, for example, to reconcile sociologists who advocate a transcending form of trust - the confidence - and economists who do not believe that confidence exists (because they are inspired by rational choice theory).
>
> Please see the new version of the paper with the revised introduction using these elements.

---

### Author Response · Authors · 2021-11-19
**Summary of the authors' answer to the reviewers**

Dear Reviewers,

We sincerely thank you for taking the time to read our paper and provide us valuable feedback. We reviewed our paper with your insightful and helpful comments and suggestions.

In revised paper, we have shortened the introduction and we have address the global perspective of our work. Having clarified this point, we have shown the importance of reviewing the definitions of trust used so far in computer science. We clarified the definition of trust, and the drivers used (brain learning, not evolution). We extended the related work with a section on trust modelling in macroeconomics. We specified and clarified the notation in the background section. We have elaborated on section "4. Method" to make the presentation of our contribution more detailed and precise : (1) We have updated and corrected the definition of trust. (2) We introduced a new section to justify the use of distributed reinforcement learning and to link our model to recent findings in neuroscience. (3) We rewrote section 4 in an effort to better justify formulae 3 and 5-7. (4) We have reordered the paragraphs to separate the presentation of the multi-agent case from the application in the single-agent case. (5) We have added a graphical representation of the algorithm. At last we recognize that additional experiments are needed to provide a strong payoff to all the setup ; they are listed in our future work. On the other hand, we hope to show the importance of new approaches like ours with the new version of the paper - as we need the assessment of the community to go further on.

---

### Decision · Program_Chairs · 2022-01-20

**Decision:**

Reject

**Comment:**

The paper presents a methodology for modeling, and learning, trust in a multi-agent reinforcement learning system. The reviewers considered this to be an interesting and important question to answer. Nevertheless, they maintained concerns on multiple fronts. The paper could benefit from being more focused. Authors are strongly encouraged to further scale down the claims in the introduction, and ensure that claims made there and later in the paper are matched with experiments that quantify, and validate, the notions introduced. Model choices made, as well assumptions introduced should be clearly motivated/mapped to reality, in light of their strength. Extending experiments to broader example settings, as outlined in the reviews, would also strengthen the work.